# Perceptions and predictors of COVID-19 vaccine hesitancy among healthcare providers across five countries in sub-Saharan Africa

Isabel Madzorera[1,☉]*, Livesy Naafoe Abokyi[2,3☉], Edward Apraku[2], Temesgen Azemraw[4], Valentin Boudo[5], Christabel James[6], Dongqing Wang[7], Frank Mapendo[8], Ourohiré Millogo[5], Nega Assefa[4], Angela Chukwu[9], Firehiwot Workneh[10], Bruno Lankoande[11], Elena C. Hemler[7], Abbas Ismail[12], Sulemana Abubakari[2], Kwaku Poku Asante[2], Yemane Berhane[10], Japhet Killewo[8], Ayoade Oduola[6], Ali Sie[5], Abdramane Soura[11], Mary Mwanyika-Sando[8], Said Vuai[12], Emily Smith[13], Till Baernighausen[14], Raji Tajudeen[15], Wafaie W. Fawzi[7,16,17]*

1 Division of Community Health Sciences, School of Public Health, University of California, Berkeley, Berkeley, California, United States of America, 2 Kintampo Health Research Center, Research and Development Division, Ghana Health Service, Kintampo, Ghana, 3 Department of Health Services, Policy, Planning, Management and Economics, School of Public Health, University for Development Studies, Tamale, Ghana, 4 College of Health and Medical Sciences, Haramaya University, Harar, Ethiopia, 5 Nouna Health Research Center, Nouna, Burkina Faso, 6 University of Ibadan Research Foundation, Ibadan, Nigeria, 7 Department of Global Health and Population, Harvard T.H. Chan School of Public Health, Harvard University, Boston, Massachusetts, United States of America, 8 Africa Academy for Public Health, Dar es Salaam, Tanzania, 9 Department of Statistics, University of Ibadan, Ibadan, Nigeria, 10 Addis Continental Institute of Public Health, Addis Ababa, Ethiopia, 11 Institut Supérieur des Sciences de la Population, University of Ouagadougou, Ouagadougou, Burkina Faso, 12 College of Natural and Mathematical Sciences, University of Dodoma, Dodoma, Tanzania, 13 Department of Global Health, Milken Institute School of Public Health, George Washington University, Washington, DC, United States of America, 14 Heidelberg Institute of Global Health, University of Heidelberg, Heidelberg, Germany, 15 Division of Public Health Institutes and Research, Africa Centres for Disease Control and Prevention, Addis Ababa, Ethiopia, 16 Department of Nutrition, Harvard T.H. Chan School of Public Health, Harvard University, Boston, Massachusetts, United States of America, 17 Department of Epidemiology, Harvard T.H. Chan School of Public Health, Harvard University, Boston, Massachusetts, United States of America

☉ These authors contributed equally to this work as first authors.
* imadzorera@berkeley.edu (IM); mina@hsph.harvard.edu (WWF)

## Abstract

The African continent has some of the world's lowest COVID-19 vaccination rates. While the limited availability of vaccines is a contributing factor, COVID-19 vaccine hesitancy among healthcare providers (HCP) is another factor that could adversely affect efforts to control infections on the continent. We sought to understand the extent of COVID-19 vaccine hesitancy among HCP, and its contributing factors in Africa. We evaluated COVID-19 vaccine hesitancy among 1,499 HCP enrolled in a cross-sectional study conducted as a telephone survey in Burkina Faso, Ethiopia, Nigeria, Tanzania, and Ghana between July to December of 2021. We defined COVID-19 vaccine hesitancy among HCP as self-reported responses of definitely not, maybe, unsure, or undecided on whether to get the COVID-19 vaccine, compared to definitely getting the vaccine. We used log-binomial or modified Poisson regression models to evaluate factors influencing vaccine hesitancy

**Data availability statement:** Individual participant data cannot be shared publicly. A data transfer agreement between Harvard T.H. Chan School of Public Health, Africa Academy for Public Health, and participating institutions (including Addis Continental Institute of Public Health, Nouna Health Research Center, Muhimbili University of Health and Allied Sciences, University of Dodoma, University of Ibadan, and Heidelberg Institute of Global Health) stipulates that data will be kept confidential and will not be shared beyond the research teams without prior permission. The de-identified dataset supporting this research may be made available following a request submitted to ghp@hsph.harvard.edu and be granted after obtaining permission from each participating institution.

**Funding:** This work was supported by institutional support from Harvard T.H. Chan School of Public Health, Boston, MA (WWF); Harvard University Center for African Studies, Boston, MA (WWF); Heidelberg Institute of Global Health, Germany (TB) and the George Washington University Milken Institute of Public Health, Washington, DC (ES). The funders had no role in study design, data collection and analysis, the decision to publish, or the preparation of the manuscript.

**Competing interests:** The authors have declared that no competing interests exist.

among HCP. Approximately 65.6% of the HCP interviewed were nurses and the mean age (±SD) of participants was 35.8 (±9.7) years. At least 67% of the HCP reported being vaccinated. COVID-19 vaccine hesitancy affected 45.7% of the HCP in Burkina Faso, 25.7% in Tanzania, 9.8% in Ethiopia, 9% in Ghana and 8.1% in Nigeria. Among unvaccinated HCP reasons for low vaccine uptake included concern about vaccine effectiveness, side effects, and fear of receiving experimental and unsafe vaccines. HCP reporting that COVID-19 vaccines are very effective (RR: 0.21, 95% CI: 0.08, 0.55), and older HCP (45 or older vs.20–29 years, RR: 0.65, 95% CI: 0.44, 0.95) were less likely to be vaccine-hesitant. Nurses were more likely to be vaccine-hesitant (RR 1.38, 95% CI: 1.01, 1.89) than doctors. Information asymmetry among HCP, beliefs about vaccine effectiveness, and the endorsement of vaccines by public health institutions may be important. Efforts to address hesitancy should consider information and knowledge gaps among different cadres of HCP alongside efforts to increase vaccine supply.

## Introduction

The Coronavirus disease 2019 (COVID-19) caused by the novel severe acute respiratory syndrome coronavirus 2 (SARS-CoV-2) continues to be a major public health challenge globally. As of March 25, 2024, there were at least 775 million cases of COVID-19 worldwide, and over 7.0 million associated deaths [1]. On the African continent, close to 9.6 million cases and 175,503 deaths had been recorded, accounting for 2.5% of all global COVID-19 related mortality in the same period [1]. COVID-19 however could continue to pose a threat to communities in sub-Saharan Africa (SSA) in the future, largely due to the lower availability of vaccines and hesitancy to accept vaccines by some in the population [2].

While other regions have made significant investments in vaccine rollout to curtail the further spread of COVID-19, efforts on the African continent continue to be affected by global inequity in access to vaccines [3]. By the end of December 2021, only 7 African countries had achieved the target of vaccinating 40% of their population [4]. While at least 68% of the global population had received at least one dose of the COVID-19 vaccine, Africa had the lowest share of the global vaccinated population with approximately 22% of the population having been fully vaccinated, and less than 1% having received a booster dose against COVID-19 [5,6]. The potential undeterred spread of the COVID-19 virus and potential future mutations have serious ramifications for a continent already dealing with significant health and economic challenges, including food insecurity, high food prices, inadequate diets, and slowed economic growth [7,8].

Vaccine hesitancy, defined as the delay or reluctance of people or communities to receive safe and recommended vaccines, predates the COVID-19 pandemic [9,10]. Apart from vaccine availability, COVID-19 vaccine hesitancy is a global threat to achieving community immunity [11]. On the African continent, only 1 in 4 healthcare providers (HCP) were fully vaccinated against COVID-19 at the time of the study [12]. The low levels of vaccination among HCP in Africa may have been due to the lower availability of COVID-19 vaccines but also partly influenced by hesitancy to take the vaccine. COVID-19 vaccine hesitancy among HCP could be a serious threat to efforts to combat the pandemic in Africa as they play an essential role in the management and control of COVID-19; and, have a high risk of getting infected. They are also a source of information on COVID-19 for the public and exert influence on public opinion in their contexts. COVID-19 vaccine hesitancy among HCP could contribute to hesitancy in the general public and increase patient risk of contracting COVID-19.

Early studies in some SSA countries indicated a high willingness to take COVID-19 vaccines, however, this was before the availability of the vaccines [9,13]. A few studies have assessed hesitancy in COVID-19 vaccine uptake among HCP in Africa [11,14,15]. A systematic review found COVID-19 acceptance rates of 46% across Africa and elevated levels of vaccine hesitancy have been reported [16,17]. However, COVID-19 acceptance rates as low as 28% have also been reported in Central Africa [16]. One meta-analysis of studies that used the health belief model to guide the understanding of determinants of COVID-19 vaccine hesitancy found that perceived barriers and perceived benefits were most consistently associated with vaccination hesitancy [18]. Other factors such as perceived susceptibility or severity, self-efficacy, and cues to action among others were informative to a lesser extent [18]. However, the review did not include studies from Africa and factors influencing hesitancy may be different in the region. Other studies suggest that factors influencing attitudes towards COVID-19 vaccines include fears about vaccine safety given the rapid development of vaccines, serious side effects, efficacy, lack of information, distrust of science, and religious reasons [16]. Gaps remain in our understanding of the extent of vaccine hesitancy among HCP in various contexts in SSA and the factors associated with it.

Understanding vaccine hesitancy and its predictors among HCP is important to inform strategies to enhance vaccination rates on the African continent [9]. This study aimed to assess the magnitude and determinants of COVID-19 vaccine hesitancy among HCP across five countries in SSA that are part of the Africa Research Implementation Science and Education (ARISE) Network, Burkina Faso, Ethiopia, Nigeria, Tanzania, and Ghana.

## Materials and methods

### Study design and study setting

The study was part of a repeated cross-sectional surveys conducted to assess knowledge and practices related to COVID-19 prevention and vaccination. The studies have evaluated the impact of COVID-19 on nutrition, health, and other domains among adolescents, adults, and HCP in 5 countries in the ARISE Network. The ARISE network is a platform for public health research and training and includes 21 member institutions across nine sub-Saharan African countries.

There were two survey rounds in the study. Briefly, in the baseline survey, we collected data from HCPs in urban areas of 3 SSA countries, Burkina Faso (Ouagadougou), Ethiopia (Addis Ababa), and Nigeria (Lagos). Information on the ARISE sites in Burkina Faso, Ethiopia, and Nigeria and their characteristics are described elsewhere [19]. In this study, we assessed COVID-19 knowledge, perceptions, preventive measures, stigma, and mental health among HCPs. The baseline survey collected data from health professionals but did not include questions on vaccine hesitancy as COVID-19 vaccines were unavailable in Africa at the time. In the round 2 survey, data were collected from the baseline sites and additionally in rural Ghana (Kintampo) and urban Tanzania (Dar es Salaam). The purpose of the Round 2 survey was to assess the impacts of prolonged exposure to the COVID-19 pandemic on nutrition, health, food security, and other factors among adults and adolescents in the selected countries, and the findings are presented in other manuscripts (21–26). Additionally, in the Round 2 surveys, we included questions about vaccine hesitancy and were able to determine its extent and contributing factors among HCP, adults and adolescents in the selected regions of SSA. Information on the design of the Round 2 surveys is described on the Harvard University Center for African Studies website (https://africa.harvard.edu/files/african-studies/files/arise_covid_survey_round_2_methods_brief_final.pdf).

Eligible participants for this study were HCP currently employed in health centers in the study areas. In the Round 1 study, HCP were recruited through lists provided by professional

associations and health facilities in each country [20]. Sampling frames for the study were developed using databases of HCP and their telephone numbers were provided by professional associations and health facilities in selected sites in each country. The study sites randomly selected 500 HCP to interview from the provided sampling lists in each country in Round 1 in urban areas of Ethiopia, Burkina Faso and Nigeria, with a target to recruit 300 HCP [19]. Due to resource and time constraints, the study was limited in the ability to select a larger sample. In the new round 2 sites of Tanzania and Kintampo, HCP were recruited from medical professional associations and healthcare facilities in urban Dar es Salaam and rural Kintampo. In all sites, the inclusion criteria were HCPs currently working in a healthcare setting, inclusive of all types of health facilities where COVID-19-related services were provided. We excluded dentists, pharmacists, and other health providers in specialties unlikely to deliver COVID-19 related medical services in the study contexts. Further, doctors and nurses are available in greater numbers and have greater interface with patients in these contexts.

In Ethiopia, Burkina Faso and Nigeria, HCP participants from Round 1 who were available were first interviewed in the Round 2 study. New participants were then randomly recruited from existing sampling frames to replace unavailable participants from Round 1 to meet study sample size requirements. The target sample size for the Round 2 survey was 300 HCP from each of the sites. We only assessed vaccination status in a few selected health centers in the study sites, therefore our estimates are not meant to be nationally representative.

This study utilized computer-assisted telephone interviews to collect data from HCP currently employed in government, public, and private health facilities in the study sites. In this analysis, we used data from the second-round survey. Data for this survey was collected between July to December of 2021. Of the 900 HCPs interviewed in Round 1, there were 548 participants retained in Round 2 of the survey indicating a retention rate of 61%. S1 Table shows the number of participants in surveys 1 and 2 of the study.

Study data were collected by trained research assistants using standardized survey questionnaires that were adapted to the sites. Research assistants collected data on socio-demographic characteristics, including age, sex, occupation of the HCP, their knowledge, attitudes, practices and perceptions of COVID-19, as well as vaccine-related beliefs and hesitancy.

## Outcome: Vaccine hesitancy

The study only assessed vaccine hesitancy at Round 2. We asked respondents if a COVID-19 vaccine was available and, if would they get it. We defined COVID-19 vaccine hesitancy among unvaccinated HCPs as responses of definitely not getting the vaccine, maybe, unsure, or undecided on whether to get the COVID-19 vaccine were it available, compared to responses of definitely getting the vaccine [10,21]. If the HCP was already vaccinated, they were classified as not vaccine-hesitant. Based on these survey responses, we created a binary variable for COVID-19 vaccine hesitancy (Yes/No).

## Statistical analysis

We hypothesized that in our study context vaccine hesitancy among HCP would be informed by their perceived susceptibility, perceptions of the severity of COVID-19, perceived barriers and perceived benefits, self-efficacy, cues to action, and their personal characteristics. We therefore considered COVID-19 service availability and HCP's exposure to or burden of COVID-19 patients, their knowledge levels about COVID-19 vaccines, their perceptions of COVID-19 risk, safety, effectiveness, COVID-19 workplace practices, HCP willingness to take COVID-19 vaccines and potential influencing factors (age, occupation, the type of health facility they work in, and their religion).

We used descriptive and inferential statistics for the analysis. Descriptive statistics used frequencies for categorical variables and means and standard deviations for continuous variables to summarize socio-demographic characteristics, perceptions around COVID-19 vaccines, workplace practices and key COVID-19 related practices in round 2 of the survey. We used log-binomial or modified Poisson regression models to evaluate associations between socio-demographic and other characteristics with vaccine hesitancy among HCPs in Round 2 of the study [22,23].

We evaluated potential factors associated with vaccine hesitancy among HCP. We considered the following possible predictors of vaccine hesitancy: age (20–29, 30–39, ≥40 years); respondent sex (female/male); occupation (doctor, nurse, other), health facility (government facility, private hospital, health outpost or other), religion (Catholic, none, Muslim, orthodox Christian, Protestant or other) (presented in Table 1).

We also considered the self-perceived risk of COVID-19 exposure (no risk, low risk, very high risk, high risk), perceived effectiveness of the COVID-19 vaccine (not effective at all, not very effective, somewhat effective, very effective), perceived safety of COVID-19 vaccines (very safe, somewhat safe, neither safe nor unsafe, not very safe, not at all safe) (presented in Fig 1A–C). We considered if COVID-19 testing is available in the facility where the HCP worked (Yes/No, free testing/paid testing). We also considered the type of COVID-19 testing available (Antigen test, PCR), having tested positive for COVID-19 previously (Yes/No),

**Table 1. Socio-demographic characteristics of HCP across 5 countries in sub-Saharan Africa.**

| Characteristics | Burkina Faso | Ethiopia | Nigeria | Tanzania | Ghana | Total |
|---|---|---|---|---|---|---|
| **N** | **300** | **277** | **312** | **310** | **300** | **1499** |
| **Female sex** | 138 (46.0) | 164 (59.2) | 217 (69.6) | 214 (69.0) | 164 (54.7) | 897 (59.8) |
| **Age of respondent in years** | | | | | | |
| 20–29 | 28 (9.3) | 119 (43.0) | 76 (24.4) | 113 (36.5) | 155 (51.7) | 491 (32.8) |
| 30–44 | 183 (61.0) | 115 (41.5) | 125 (40.1) | 150 (48.4) | 139 (46.3) | 712 (47.5) |
| ≥45 | 89 (29.7) | 43 (15.5) | 111 (35.6) | 47 (15.2) | 6 (2.0) | 296 (19.8) |
| Age of respondent (Mean ± SD) years | 39.9 ± 9.1 | 33.7 ± 9.7 | 39.7 ± 11.7 | 34.6 ± 8.6 | 30.8 ± 4.7 | 35.8 ± 9.7 |
| **Health care provider** | | | | | | |
| Doctor | 85 (28.3) | 118 (42.6) | 99 (31.7) | 105 (33.9) | 3 (1.0) | 410 (27.4) |
| Nurse | 197 (65.7) | 157 (56.7) | 182 (58.3) | 179 (57.7) | 269 (89.7) | 984 (65.6) |
| Clinical officer | 2 (0.7) | 2 (0.7) | 9 (2.9) | 25 (8.1) | 6 (2.0) | 44 (2.9) |
| Community health worker | 8 (2.7) | 0 (0) | 13 (4.2) | 0 (0) | 6 (2.0) | 27 (1.8) |
| Medical Assistant | 8 (2.7) | 0 (0) | 9 (2.9) | 1 (0.3) | 16 (5.3) | 34 (2.3) |
| **Health facility type** | | | | | | |
| Government hospital/clinic | 208 (69.3) | 207 (74.7) | 224 (71.8) | 182 (58.7) | 169 (56.3) | 990 (66.0) |
| Private hospital/clinic | 87 (29.0) | 51 (18.4) | 80 (25.6) | 0 (0) | 12 (4.0) | 230 (15.3) |
| Health outpost/CHPS compound) | 1 (0.3) | 16 (5.8) | 5 (1.6) | 128 (41.3) | 46 (15.3) | 196 (13.1) |
| Mission hospital | 4 (1.3) | 3 (1.1) | 3 (1.0) | 0 (0) | 73 (24.3) | 83 (5.5) |
| **Religion** | | | | | | |
| None | 1 (0.3) | 2 (0.7) | 4 (1.3) | 1 (0.3) | 0 (0) | 8 (0.5) |
| Catholic | 149 (49.8) | 3 (1.1) | 23 (7.4) | 79 (25.6) | 70 (23.3) | 324 (21.7) |
| Muslim | 106 (35.5) | 21 (7.6) | 41 (13.3) | 78 (25.2) | 40 (13.3) | 286 (19.1) |
| Orthodox Christian | 3 (1.0) | 198 (71.5) | 141 (45.6) | 1 (0.3) | 37 (12.3) | 380 (25.4) |
| Protestant and other Christian | 40 (13.4) | 53 (19.1) | 100 (32.4) | 150 (48.5) | 153 (51.0) | 496 (33.2) |

Data are shown as mean ±SD or n (percent). Acronyms: CHPS Community Health Planning and Services.

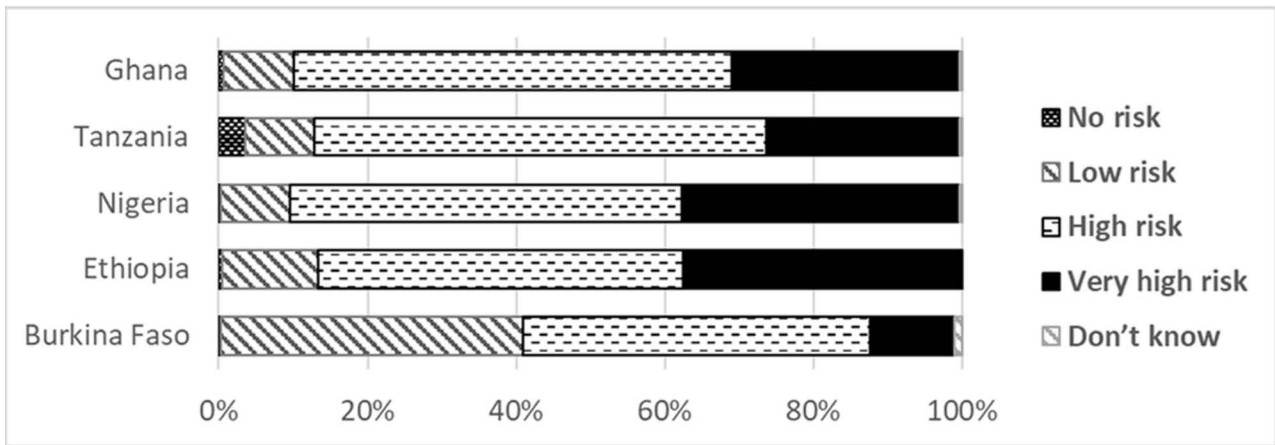

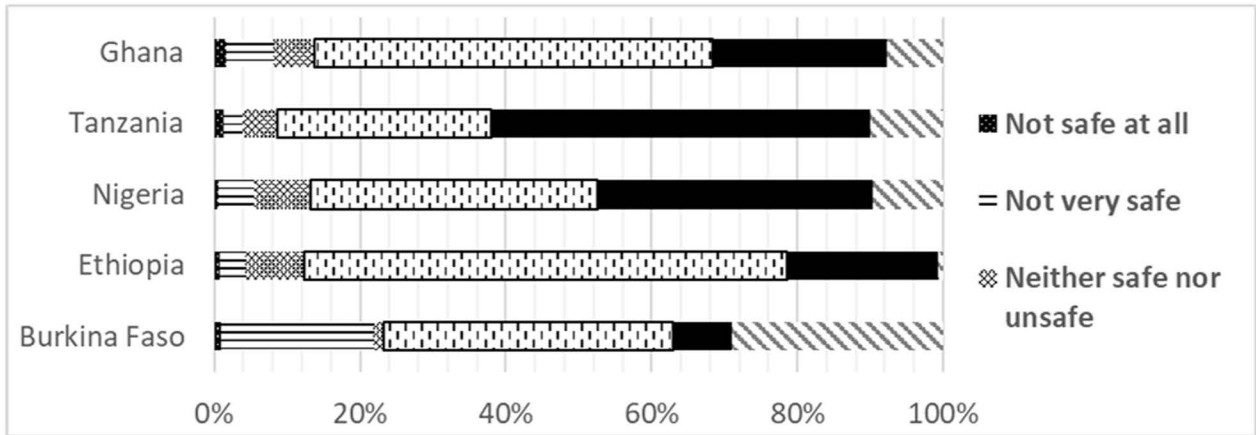

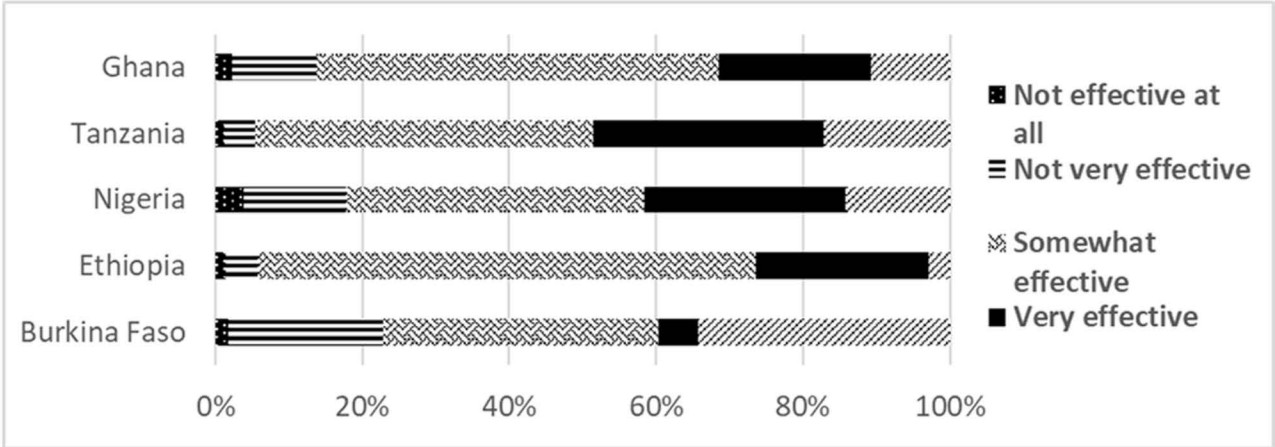

**Fig 1. A: Healthcare provider perceptions of risk of COVID-19 exposure in 5 countries in sub-Saharan Africa, B: Healthcare provider perceptions on COVID-19 vaccine safety in 5 countries in sub-Saharan Africa, C: Healthcare provider perceptions on COVID-19 vaccine effectiveness in 5 countries in sub-Saharan Africa.**

having cared for COVID-19 patients previously (never, yes and in the past one month, yes but over one month ago), workplace COVID-19 polices (Yes/No), and COVID 19 control practices (score indicating the level of prevention measures being implemented in the workplace including wearing masks, using personal protective equipment (PPE), hand washing with water and soap, social distancing, sanitizers or hand washing station in health facility, cleaning and decontamination or disinfection of public areas, checking high temperatures, Yes/No) (presented in Table 2). We also considered influence of vaccine production in Africa on willingness to take vaccine (No, will not change my mind/Yes, will decrease my chances of taking it/Yes, will increase my chances of taking it), believe COVID-19 vaccine is bioweapon (Yes/No) and World Health Organization (WHO)/UNICEF endorsement of COVID-19 vaccine as safe and effective affect likelihood get the vaccine (much more likely, more likely, no difference, less likely and much less likely).

Further, we also considered a score of reasons for not getting the COVID-19 vaccine (Do not think it is needed, not at risk of getting COVID, vaccine not effective against COVID-19, negative media reports, vaccine not safe/ developed too fast, concerned about side effects, fear of experimental vaccine, will get worse quality vaccines, fear getting COVID-19 disease from the vaccine, illnesses/autism from the vaccine, will cause infertility/sterilization/population control, religious reasons/church, microchipping fears, New World Order, bad reaction with previous vaccinations, chronic condition, e.g., diabetes, hypertension and personal liberty/do not want bodily intrusion) as a potential confounder (presented in Fig 2).

Covariates were selected for inclusion in the main model using univariate tests at $p < 0.20$. We evaluated for significant associations in the adjusted models based on a significance level of $p < 0.05$. We used the missing indicator approach to account for missing covariate data. Analysis was conducted using SAS 9.4 (Cary, NC, USA).

## Ethical approval and consent

Verbal consent was obtained from study participants before they were admitted into the study. Ethical approval for the study was obtained from the Institutional Review Board at Harvard

**Table 2. Frequency of COVID-19 service availability and knowledge about vaccines among HCP preventive practices, across 5 countries in sub-Saharan Africa.**

| COVID-19 preventive practices and beliefs about vaccines | Burkina Faso | Ethiopia | Nigeria | Tanzania | Ghana | Total |
|---|---|---|---|---|---|---|
| **COVID-19 testing and patient care in facilities** | | | | | | |
| Ever tested for COVID-19 | 142 (47.7) | 215 (77.6) | 168 (53.9) | 128 (42.7) | 42 (13.6) | 695 (46.4) |
| Ever tested positive for COVID-19 | 14 (9.8) | 68 (31.5) | 39 (23.2) | 13 (33.3) | 26 (21.7) | 160 (23.3) |
| **Ever cared for COVID-19 patients** | | | | | | |
| Never | 197 (66.1) | 44 (15.9) | 164 (53.3) | 158 (52.0) | 194 (70.3) | 757 (51.8) |
| Yes, in the past one month | 13 (4.4) | 124 (44.9) | 51 (16.6) | 7 (2.3) | 32 (11.6) | 227 (15.5) |
| Yes, over one month ago | 88 (29.5) | 108 (39.1) | 93 (30.2) | 139 (45.7) | 50 (18.1) | 478 (32.7) |
| **COVID-19 Prevention practices (Number and Proportion reporting response)** | | | | | | |
| Wearing masks | 295 (98.3) | 275 (99.3) | 311 (100) | 279 (90.3) | 298 (99.3) | 1458 (97.4) |
| Use personal protective equipment (PPE) such as gowns, goggles, and shields. | 274 (91.6) | 231 (83.7) | 231 (74.8) | 192 (62.3) | 165 (55.2) | 1093 (73.3) |
| Hand washing with water and soap | 296 (98.7) | 258 (93.1) | 309 (99.4) | 299 (96.5) | 300 (100) | 1462 (97.6) |
| Keeping sufficient distance between patients in the waiting area | 131 (44.4) | 183 (66.3) | 280 (90.3) | 203 (65.7) | 243 (81.5) | 1040 (69.9) |
| Presence of sanitizers or hand washing stations in all service delivery points | 268 (89.9) | 208 (75.6) | 301 (97.1) | 303 (97.7) | 293 (98.0) | 1373 (92.0) |
| Regular cleaning and decontamination/disinfection of public areas and offices | 268 (89.3) | 220 (80.3) | 286 (93.2) | 241 (77.7) | 292 (97.3) | 1307 (87.7) |
| Temperature measurement (high temperatures 37 degrees Celsius) | 257 (86.0) | 154 (55.6) | 279 (89.7) | 208 (67.1) | 288 (96.0) | 1186 (79.2) |

Data are shown as n (percent).

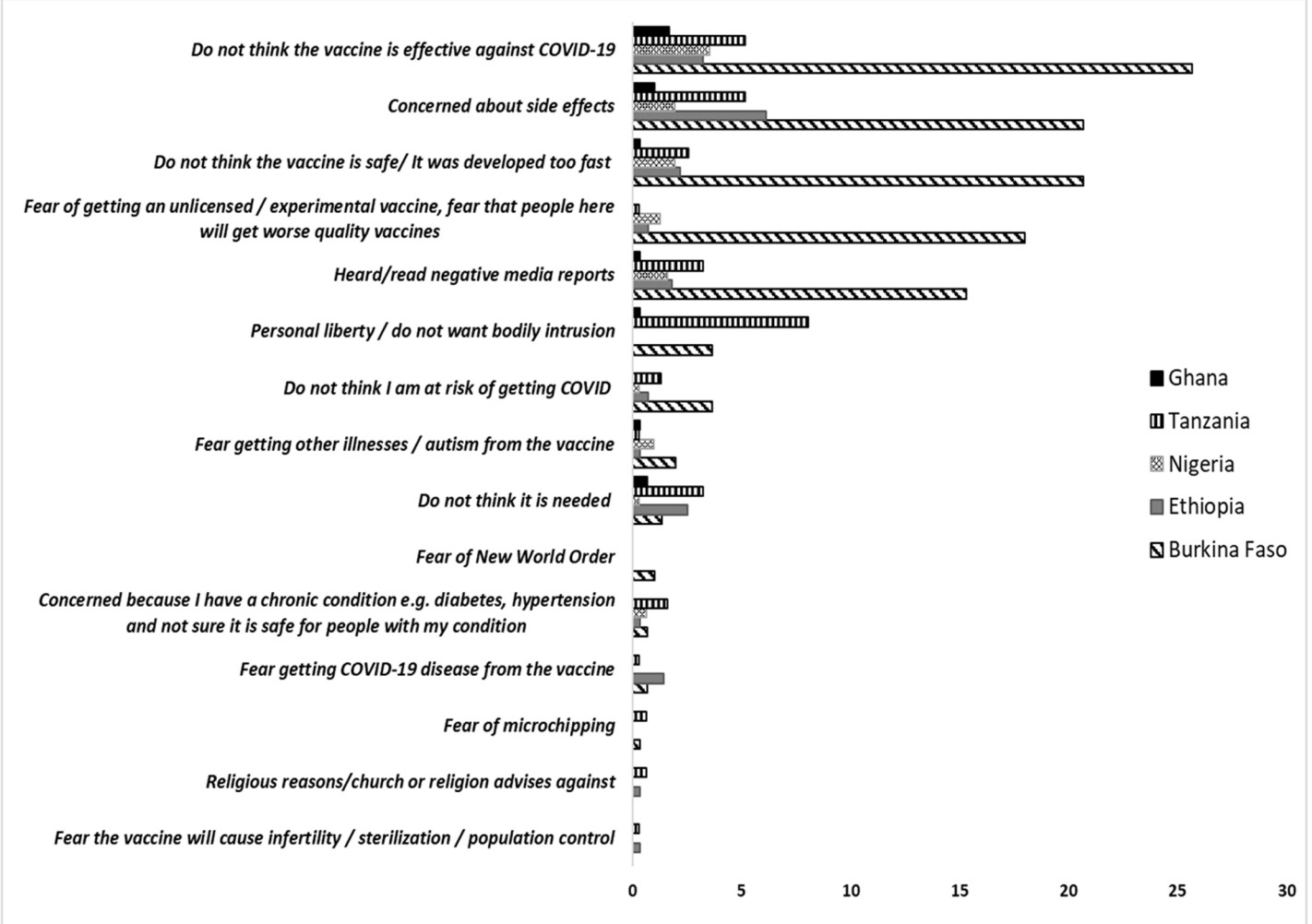

**Fig 2. Frequent reasons (%) for not getting COVID-19 vaccination in 5 countries in sub-Saharan Africa.**

T.H. Chan School of Public Health and ethical review boards in each country and site, including the Kintampo Health Research Centre Institutional Ethics Committee in Ghana; the Nouna Health Research Center Ethical Committee and National Ethics Committee in Burkina Faso; the Institutional Ethical Review Board of Addis Continental Institute of Public Health in Ethiopia; the University of Ibadan Research Ethics Committee and National Health Research Ethics Committee in Nigeria; and the Muhimbili University of Health and Allied Sciences and National Institute for Medical Research in Tanzania.

## Results

### Sociodemographic characteristics

There were 1499 HCP included in the study across the 5 SSA countries in the round 2 study. Of the participants, 300 were from Burkina Faso, 277 were from Ethiopia, 312 were from Nigeria, 310 were from Tanzania and 300 were from Ghana. Table 1 shows the socio-demographic characteristics of the study population. The majority of respondents (59.8%) across all countries were female. However, In Burkina Faso, female participants were fewer (46%) than men.

Respondents' mean age (±SD) was 35.8 (±9.7) years. Most HCP in Ghana and Ethiopia were under 30 years of age, while in all other countries, most respondents were in the 30–44 years age group. The majority of HCP assessed across all the countries were nurses (66%), with Ghana (90%) and Burkina Faso (66%) accounting for the highest proportion of nurses. The majority of respondents worked in government hospitals or clinics. In Tanzania, 41% of HCPs worked in health posts where primary health care services were provided; in Ghana, 24% worked in Mission hospitals. In Ethiopia and Nigeria, most of the respondents were Orthodox Christians. In Burkina Faso half of the respondents were Catholic, and in Ghana and Tanzania, at least 48% were Protestant or from other Christian denominations.

## COVID-19 service availability and care for patients

Testing services were available for 85% of the respondents in Ethiopia, 67% in Nigeria, 59% in Burkina Faso, 40% in Ghana, and 38% in Tanzania (results not shown). Among those facilities that had services, COVID-19 testing was free for the majority of respondents in Burkina Faso (98%), Ethiopia (90%), Nigeria (81%) and Ghana (98%) (results not shown). COVID-19 testing was paid for 60% of the time in Tanzania, 19% in Nigeria, and less than 10% of the time in other countries. Availability of PCR and antigen testing for COVID-19 was comparable, with respondents reporting 28.6% availability for the former compared to 25.4% for the latter (results also not shown). Among HCP who had ever tested for COVID-19, 23.3% had tested positive (Table 2). Almost half of the HCP in the study had never cared for COVID-19 patients, and 15.5% had cared for patients in the previous month before the survey.

## COVID-19 workplace practices

Table 2 shows prevention measures implemented in the workplace to control COVID-19. The measures that most HCPs reported as being implemented in the workplace were wearing a mask, handwashing with water and soap, and regular cleaning or decontamination of public areas (at least 92%). The least commonly practiced measures were wearing PPE (73%) and socially distancing patients in waiting rooms (70%). The use of personal protective equipment (PPE) was least reported in Ghana and Tanzania. In Burkina Faso, socially distancing patients was reported by only 44.4% of HCP, and by 65.7% in Tanzania. In Ethiopia, temperature checks were reported by only 55.6%, and in Tanzania, it was reported by 67.1%.

## Knowledge about the COVID-19 vaccines

In terms of knowledge about the COVID-19 vaccine, 88% of the HCP indicated that the side effects of the COVID-19 vaccine are usually mild (Table 3). About two-thirds of the respondents across countries reported that the vaccine was developed too fast. Additionally, half of the HCPs in all sites did not believe there was sufficient evidence that the COVID-19 vaccine prevents the occurrence and spread of COVID-19, with more than 60% of HCP in Burkina Faso and Ghana reporting this. About 17% of HCP believed that people have died after taking the COVID-19 vaccine (>22% in Ethiopia and Nigeria). Across all countries, less than 10% of respondents believed that it is not necessary to get a COVID-19 vaccine if one follows all COVID-19 safety protocols (20% in Burkina Faso and 12% in Ethiopia). Less than 11% of respondents believed COVID-19 is a conspiracy or a bioweapon and 4% reported that people on the African continent are immune to COVID-19.

## Perceptions of COVID-19 risk, safety, effectiveness

Almost all HCP from study countries perceived a high to very high risk of exposure to COVID-19 except in Burkina Faso, where about 41% of HCP perceived a low risk of exposure

**Table 3. Knowledge about COVID-19 vaccine among Health Care Providers across 5 countries in sub-Saharan Africa.**

| Knowledge about COVID-19 vaccine among Health Care Providers | Burkina Faso | Ethiopia | Nigeria | Tanzania | Ghana | Total |
|---|---|---|---|---|---|---|
| **Knowledge about COVID-19 vaccines (Number and Proportion reporting response)** | | | | | | |
| Vaccine trial participants have died after taking the vaccine | 37 (19.6) | 51 (22.4) | 53 (23.7) | 20 (9.5) | 26 (10.6) | 187 (17.1) |
| Side effects from the COVID-19 vaccine are usually mild and temporary and should go away in a few days | 253 (94.4) | 223 (80.8) | 261 (86.4) | 216 (83.4) | 278 (95.5) | 1231 (88.2) |
| There is no need for a vaccine because COVID-19 is a conspiracy or a bioweapon | 26 (10.4) | 16 (6.1) | 10 (3.4) | 8 (2.9) | 12 (4.3) | 72 (5.3) |
| People on the African continent are immune to COVID-19, so there is no need for a vaccine | 13 (4.9) | 17 (6.2) | 5 (1.7) | 10 (3.4) | 12 (4.1) | 57 (4.0) |
| It is not necessary to get a COVID-19 vaccine if you follow all safety protocols | 56 (19.7) | 32 (11.6) | 16 (5.3) | 10 (3.3) | 23 (7.8) | 137 (9.4) |
| There is not enough evidence that the COVID-19 vaccine prevents the occurrence and spread of COVID-19 | 166 (65.6) | 89 (32.7) | 143 (49.3) | 139 (52.9) | 168 (61.8) | 705 (52.2) |
| The COVID-19 vaccine was developed too fast. | 240 (87.3) | 141 (53.0) | 132 (49.4) | 234 (82.1) | 160 (60.4) | 907 (66.8) |

Data are shown as n (percent).

(Fig 1A). At least 28% of all HCP believed that the COVID-19 vaccines were very safe and up to 45% thought they were somewhat safe (Fig 1B). The majority of respondents believed that COVID-19 vaccines were very or somewhat effective in preventing the disease. Only 1% of the HCP believed the COVID-19 vaccines were not safe at all; and 8% believed they were not very safe, with the highest proportion in Ghana (21.0%). In Ethiopia, 32% of respondents believed vaccines were not very effective (Fig 1C).

## HCP willingness to take COVID-19 vaccines

Table 4 shows beliefs about the HCP's willingness to take COVID-19 vaccines. At least 85% of the HCP indicated that COVID-19 vaccines were available in their country and localities, except in Ghana, where only 45.7% reported vaccine availability at the time of the interview. The majority of HCPs (84%) indicated that the COVID-19 vaccine was available to them as HCPs. At least 68% of HCP had been vaccinated by the time of the survey, and 11% expected to be vaccinated by the end of 2021. Vaccination rates were lowest in Burkina Faso (40.3%), Tanzania (66.5%) and Ghana (69.3%). At least 86% of HCP indicated that it was important for HCPs to get vaccinated and at least 79% reported that their workplaces had formulated COVID-19 policies. Most participants from Nigeria, Ghana and Ethiopia reported having workplace guidelines on COVID-19 but only 61.6% and 68% from Burkina Faso and Tanzania, respectively, had guidelines.

At least 26% of HCPs indicated that the origin of the vaccine influenced their willingness to take COVID-19 vaccines. A similar proportion of respondents indicated that if a vaccine was developed and tested in Africa, this would increase their willingness to take it. In Burkina Faso, up to 67% of HCP indicated a greater willingness to take African vaccines.

The majority of unvaccinated HCP indicated a willingness to take the vaccine if it was availed to them (S1 Fig). However, in Tanzania, 48.1% of unvaccinated HCPs reported unwillingness and 18.3% were undecided on whether to take the COVID-19 vaccine. Reasons for unwillingness to take the COVID-19 vaccines across the countries include perceptions the vaccine was not effective, concern about side effects or that vaccine development was too fast, fear of receiving poor quality vaccines and negative media reports. These concerns were most

**Table 4. Perceptions of the safety, availability of vaccines and willingness to get the COVID-19 vaccine and expected benefits in 5 countries in sub-Saharan Africa.**

| Characteristics | Burkina Faso | Ethiopia | Nigeria | Tanzania | Ghana | Total |
|---|---|---|---|---|---|---|
| **The perceived safety of vaccines** | | | | | | |
| In general, I believe that vaccines are safe | 153 (75.7) | 210 (76.6) | 292 (99.0) | 250 (89.3) | 279 (95.6) | 1184 (88.2) |
| **Vaccine hesitant** | 137 (45.7) | 27 (9.8) | 25 (8.1) | 79 (25.7) | 27 (9.0) | 295 (19.7) |
| **Availability of COVID-19 vaccines** | | | | | | |
| COVID-19 vaccine available in the country | 298 (100) | 237 (85.6) | 306 (98.1) | 306 (98.7) | 137 (45.7) | 1284 (85.8) |
| COVID-19 vaccine has been made available to you as a healthcare provider | 192 (64.4) | 258 (93.1) | 287 (92.0) | 299 (96.5) | 225 (75.0) | 1261 (84.2) |
| **Have you or any of your colleagues received the COVID-19 vaccination** | | | | | | |
| Yes, I have been vaccinated | 120 (40.0) | 226 (81.6) | 257 (82.4) | 206 (66.5) | 208 (69.3) | 1017 (67.9) |
| I know someone who has been vaccinated | 135 (45.0) | 29 (10.5) | 27 (8.7) | 97 (31.3) | 81 (27.0) | 369 (24.6) |
| **Workplace has formulated policies or guidelines related to COVID-19** | 180 (61.6) | 208 (76.5) | 297 (95.8) | 208 (68.0) | 269 (92.8) | 1162 (79.1) |
| **How important is it for Healthcare providers to get vaccinated against COVID-19** | | | | | | |
| Very important | 211 (71.5) | 217 (78.6) | 291 (94.5) | 284 (92.8) | 288 (96.3) | 1291 (87.0) |
| Somewhat important | 69 (23.4) | 49 (17.8) | 15 (4.9) | 19 (6.2) | 10 (3.3) | 162 (10.9) |
| Not very important | 11 (3.7) | 4 (1.5) | 2 (0.7) | 3 (1.0) | 0 | 20 (1.4) |
| Not important at all | 4 (1.4) | 6 (2.2) | 0 | 0 | 1 (0.3) | 11 (0.7) |
| **COVID-19 vaccine's country of origin affects my willingness to take the vaccine** | 74 (24.9) | 107 (38.6) | 56 (18.0) | 68 (21.9) | 91 (30.3) | 396 (26.5) |
| **If COVID-19 vaccine had been developed or tested in Africa would that affect your willingness to take it or recommend it** | | | | | | |
| No, will not change my mind | 89 (30.7) | 191 (71.0) | 198 (67.1) | 205 (70.5) | 209 (70.9) | 892 (61.9) |
| Yes, will decrease my chances of taking it | 2 (0.7) | 46 (17.1) | 41 (13.9) | 18 (6.2) | 26 (8.8) | 133 (9.2) |
| Yes, will increase my chances of taking it | 199 (68.6) | 32 (11.9) | 56 (19.0) | 68 (23.4) | 60 (20.3) | 415 (28.8) |
| **When do you think a COVID-19 vaccine will be made available to you** | | | | | | |
| Never | 4 (1.3) | 20 (7.2) | 3 (1.0) | 16 (5.2) | 2 (0.7) | 45 (3.0) |
| Already received the vaccine | 119 (39.8) | 226 (81.6) | 257 (82.4) | 206 (66.5) | 208 (69.3) | 1016 (67.8) |
| Before the end of 2021 | 88 (29.4) | 15 (5.4) | 21 (6.7) | 12 (3.9) | 26 (8.7) | 162 (10.8) |
| During the first six months of 2022 | 8 (2.7) | 9 (3.3) | 3 (1.0) | 7 (2.3) | 9 (3.0) | 35 (2.4) |
| During the last six months of 2022 | 2 (0.7) | 1 (0.4) | 3 (1.0) | 4 (1.3) | 2 (0.7) | 12 (0.8) |
| 2023 or later | 38 (12.5) | 4 (1.4) | 2 (0.6) | 9 (2.9) | 9 (3.0) | 62 (4.1) |
| Don't know | 39 (13.0) | 2 (0.7) | 21 (6.7) | 50 (16.1) | 44 (14.7) | 156 (10.4) |
| **Would not recommend that my friends and loved ones get the COVID-19 vaccine** | 57 (19.1) | 26 (9.4) | 11 (3.5) | 33 (10.7) | 6 (2.0) | 133 (8.9) |
| **Are you doing any activities to boost your community's confidence in taking the vaccine** | | | | | | |
| Community outreach/education | 216 (72.2) | 228 (82.3) | 185 (59.3) | 261 (84.2) | 193 (64.3) | 1083 (72.3) |
| Other | 0 (0) | 0 (0) | 23 (7.4) | 0 (0) | 9 (3.0) | 32 (2.2) |
| **Willing to participate in a vaccine clinical trial if available** | 181 (60.5) | 136 (49.1) | 102 (32.7) | 69 (22.3) | 141 (47.0) | 629 (42.1) |

Data are shown as n (percent).

frequently reported in Burkina Faso (Fig 2). HCPs reported that the most prescribed treatments for COVID-19 were antibiotics/azithromycin (47.2%) and multivitamins (33.4%) (S2 Fig). Close to 20% of the cases were not prescribed any medications.

## Potential determinants of COVID-19 vaccine hesitancy

COVID-19 vaccine hesitancy affected 19.7% of the HCPs (Table 3). Reported hesitancy was highest in Burkina Faso (45.7%) and Tanzania (25.7%) compared to other countries. The lowest levels of vaccine hesitancy were reported in Nigeria at 8.1% and Ghana at 9.0%.

Table 5 shows factors associated with COVID-19 vaccine hesitancy among HCP in the study. We found that there were site-specific differences in the risk of COVID-19 vaccine

**Table 5. Factors associated with COVID-19 vaccine hesitancy among healthcare providers in round 2 of the ARISE COVID study in 5 countries.**

| | Unadjusted | Adjusted |
|---|---|---|
| | RR (95% CI) | RR (95% CI) |
| **Country** | | |
| Burkina Faso | Ref | ref |
| Ethiopia | 0.21 (0.14, 0.32)*** | 0.52 (0.29, 0.95)* |
| Nigeria | 0.18 (0.12, 0.27)*** | 0.67 (0.37, 1.23) |
| Tanzania | 0.56 (0.43, 0.74)*** | 1.49 (0.99, 2.25) |
| Ghana | 0.56 (0.43, 0.74)*** | 0.65 (0.36, 1.17) |
| **Male sex** | 0.86 (0.68, 1.09) | 0.90 (0.69, 1.18) |
| **Age** | | |
| <30 years | Ref | ref |
| 30–44 years | 1.19 (0.92, 1.54) | 0.80 (0.59, 1.08) |
| 45 years+ | 0.97 (0.69, 1.36) | 0.65 (0.44, 0.95)* |
| **Occupation** | | |
| Doctor | Ref | ref |
| Nurse | 1.18 (0.90, 1.54) | 1.38 (1.01, 1.89)* |
| Other occupation | 1.03 (0.62, 1.70) | 1.52 (0.88, 2.63) |
| **Religion** | | |
| Catholic or none | Ref | ref |
| Muslim | 3.10 (2.10, 4.60)*** | 1.00 (0.58, 1.72) |
| Orthodox Christian | 3.18 (2.13, 4.74)*** | 1.16 (0.68, 1.99) |
| Protestant or other | 1.98 (1.33, 2.95)** | 1.16 (0.71, 1.89) |
| **Health center type** | | |
| Government hospital/clinic | Ref | ref |
| Private hospital/clinic | 1.12 (0.42, 1.51)* | 0.96 (0.70, 1.33) |
| Health outpost/CHPS compounds or other | 0.58 (0.41, 0.84)** | 0.73 (0.47, 1.13) |
| **Perceived risk of COVID-19 disease** | | |
| Low risk | 1.30 (0.48, 3.53) | 0.94 (0.33, 2.69) |
| No risk | 0.65 (0.24, 1.75) | 0.93 (0.33, 2.61) |
| Very high risk | 0.58 (0.21, 1.59) | 1.08 (0.38, 3.06) |
| High risk | Ref | ref |
| **Effectiveness of COVID-19 vaccine** | | |
| Very effective | 0.05 (0.02, 0.12)*** | 0.21 (0.08, 0.55)* |
| Somewhat effective | 0.31 (0.18, 0.53)*** | 0.87 (0.46, 1.63) |
| Not very effective | 0.91 (0.52, 1.58) | 0.97 (0.53, 1.80) |
| Not effective at all | Ref | ref |
| **Workplace policy score (number of policies)** | 0.75 (0.68, 0.82)*** | 0.98 (0.88, 1.09) |
| **Reasons for not vaccinating (number of reasons)** | 2.03 (1.91, 2.16)*** | 1.78 (1.62, 1.95)*** |
| **Believe the COVID-19 vaccine is a bioweapon** | 2.89 (1.99, 4.20)*** | 0.94 (0.62, 1.42) |
| **Believe vaccines are safe** | 0.30 (0.22, 0.40)*** | 0.71 (0.50, 1.02) |
| **WHO/UNICEF's endorsement of the COVID-19 vaccine as safe and effective affects the likelihood to get the vaccine** | | |
| Much more likely | 0.19 (0.13, 0.26)*** | 0.51 (0.35, 0.74)*** |
| More likely | 0.40 (0.31, 0.52)*** | 0.69 (0.52, 0.92)* |
| No difference | Ref | ref |

*(Continued)*

**Table 5.** (Continued)

|  | Unadjusted | Adjusted |
|---|---|---|
|  | RR (95% CI) | RR (95% CI) |
| Less likely | 0.50 (0.27, 0.90)* | 0.92 (0.47, 1.83) |
| Much less likely | 0.84 (0.31, 2.29) | 1.53 (0.51, 4.56) |

Acronyms: COVID-19; Coronavirus disease 2019, RR: relative risk, CI: Confidence interval, * <0.05, ** <0.01, *** <0.001. Variables included in the final model include country, age, respondent sex, occupation (doctor, nurse, other), health facility (government facility, private hospital, health outpost or other), religion (Catholic, none, Muslim, orthodox Christian, Protestant or other), perceived risk of COVID-19 exposure (no risk, low risk, very high risk, high risk), perceived effectiveness of the COVID-19 vaccine (not effective at all, not very effective, somewhat effective, very effective), perceived safety of COVID-19 vaccines (yes/no), COVID-19 vaccine is bioweapon (Yes/No), COVID 19 control practices (score of prevention measures being implemented in workplace (wearing masks, using personal protective equipment (PPE), hand washing with water and soap, social distancing, sanitizers or hand washing station in health facility, cleaning and decontamination or disinfection of public areas, checking temperature), yes/no), number of reasons for not vaccinating, WHO/UNICEF endorsement of COVID-19 vaccine as safe and effective affect vaccination (much more likely, more likely, no difference, less likely and much less likely).

hesitancy among HCP. In Ethiopia (RR: 0.52, 95% CI: 0.29, 0.95) HCP had lower vaccine hesitancy, and in Tanzania, there was a trend towards having higher vaccine hesitancy (RR: 1.49, 95% CI: 0.99, 2.25, p = 0.06) among HCP compared to Burkina Faso. Age was associated with vaccine hesitancy, with HCPs aged 45 years or older less likely to be vaccine-hesitant (RR: 0.65, 95% CI: 0.44, 0.95) compared to those aged 20–29 years. Nurses were more likely to be vaccine-hesitant (RR 1.38, 95% CI: 1.01, 1.89) compared to doctors. Respondents reporting that COVID-19 vaccines are very effective (RR: 0.21, 95% CI: 0.08, 0.55) were less likely to report vaccine hesitancy. For each reason given by HCP for not wishing to receive COVID-19 vaccination, the risk of being vaccine-hesitant increased (RR 1.78, 95% CI: 1.62, 1.95). Finally, HCP reporting that the World Health Organization or UNICEF endorsement of the COVID-19 vaccine as safe and effective would make them more likely (RR 0.69, 95% CI: 0.52, 0.92) or much more likely (RR 0.51, 95% CI: 0.35, 0.74) to receive the vaccine were less likely to be vaccine-hesitant.

As a sensitivity analysis we reclassified hesitancy, excluding vaccine refusal cases. When we analyzed factors associated with hesitancy we found that country, vaccine effectiveness, having numerous reasons for not vaccinating and WHO/UNICEF endorsement of vaccine influence were significant predictors of hesitancy (results not shown).

## Discussion

We assessed factors influencing COVID-19 vaccine hesitancy among HCP in Burkina Faso, Ethiopia, Ghana, Nigeria and Tanzania between July to December of 2021. We found that the majority of HCP had been vaccinated by the time of the survey. Almost a fifth of the HCPs across the sites were hesitant to receive the COVID-19 vaccine. There were differences in vaccine hesitancy by the site. The age of a HCP, their profession and perceptions about COVID-19 vaccine effectiveness were significant predictors of vaccine hesitancy. Older HCP reported lower levels of vaccine hesitancy, while nurses and HCP who believed that COVID-19 vaccines were not effective were more likely to be vaccine-hesitant.

Our findings show consistency with some findings from previous studies. In a study among HCPs in South Africa, being an older HCP or a physician and perceptions of the benefits and risks of vaccination were associated with lower vaccine hesitancy [13]. However, the study also found that beliefs that vaccines are incompatible with religion and willing-ness to be vaccinated to protect others informed hesitancy [13]. In Ethiopia, older age (>40

years) and being a medical doctor were associated with a lower risk of vaccine hesitancy and in Nigeria HCP who were younger, single and with lower income expressed more vaccine hesitancy while being a nurse or doctor were associated with lower hesitancy [17]. In Ghana, being female and having a low income have previously also been associated with increased COVID-19 vaccine hesitancy [18]. These findings suggest there may be information asymmetry among different cadres of HCPs and lower perceived risks among some population sub-groups [24]. Women additionally face greater concerns about the effects of COVID-19 vaccines on their fertility [25]. In this study, we did not observe greater hesitancy among female HCP.

In this study, we found that nurses are among the most hesitant. This is concerning given that nurses are on the front line and during the COVID-19 pandemic they had extensive contact with infected people receiving care at all levels of health systems. Vaccine hesitancy and refusal among nurses have been noted across multiple contexts [26]. Some studies have reported that factors contributing to vaccine hesitancy among nurses include religious reasons, concerns about vaccine effectiveness, potential side effects of COVID-19 vaccines, ethical concerns, and nurses with inaccurate knowledge about COVID-19 vaccines were likely to be hesitant, along with younger nurses [27,28]. Our findings and those of others suggest that understanding the causes of vaccine hesitancy within this group and addressing them will be critical in future pandemics.

Older age may be associated with lower vaccine hesitancy since age is a risk factor for COVID-19 infection, and older people are more vulnerable to more severe disease possibly due to more co-morbidities. This could be a factor influencing decisions by older HCP to be vaccinated. In an Africa-wide survey, respondents indicated that vaccine hesitancy was lower among those with a higher perceived risk of infection [19]. Additionally, HCP who perceive that vaccines are effective may face fewer personal barriers and may be more willing to take the vaccine. Other factors reported by studies include concerns about vaccine safety, serious side effects and efficacy of vaccines, with limited information also being a key factor [16]. Additionally, perceptions of the lack of COVID-19 vaccine benefits, distrust of the government and the ability of science to provide safe and effective vaccines, and concerns about vaccine safety have been associated with greater vaccine hesitancy [14].

In our study, there were differences in vaccine hesitancy among HCP by country, with Burkina Faso and Tanzania reporting the highest levels of hesitancy and Ghana and Nigeria showing lower levels. Previous studies have also shown variations in acceptance rates from as low as 28% in Central Africa, and up to 48–49% in West and East Africa [16], suggesting that there may be context-specific factors influencing attitudes towards COVID-19 vaccines. Country-specific factors such as differences in availability and access to vaccines, the burden of COVID-19 cases, and variations in mitigation measures put in place could account for these differences. Reported rates of vaccine hesitancy have been higher in other studies. A study conducted early in the COVID-19 emergency in Nigeria found higher rates of vaccine hesitancy among HCPs (50%) [17]. In a South African study, 41.0% of HCPs were vaccine-hesitant [13] and in Ghana, three out of five HCP were hesitant early in the pandemic [29]. Levels of vaccine hesitancy of 41%–45.9% have also been reported among HCP in South Africa and Ethiopia [7]. These studies were conducted early during the COVID-19 pandemic before vaccines had been available. Additionally, low rates of COVID-19 vaccine acceptance by the general population on the African continent have been reported [20]. Our observations of lower levels of vaccine hesitancy may be partly influenced by changing attitudes and beliefs towards the COVID-19 vaccine across the continent as the pandemic has progressed, due to increased cases and mortality, increased information and greater availability of vaccines.

We observed higher levels of COVID-19 vaccination among HCP across our study, compared to previous reports on the African continent. Two-thirds of HCP interviewed across the 5 countries had already received the COVID-19 vaccination at the time of the survey; however, vaccination rates were low in Burkina Faso. HCP were prioritized to receive the COVID-19 vaccine initially when the vaccines were not readily available. It is therefore expected that more HCP had received the vaccine. The WHO, however, reported that approximately 25% of HCPs were fully vaccinated against COVID-19 on the African continent by November 2021 [8]. The differences observed between this study and previous studies could be due to the fact our study is more recent, and vaccines may have been relatively more available during our study. Further, four of our survey sites were in urban areas, where vaccination rates were generally higher. Reasons for lower vaccination in some of our study sites could have been partially influenced by the low availability of the COVID-19 vaccines. This was not possible to evaluate in our study due to data unavailability. Reasons for the limited availability of COVID-19 vaccines on the African continent have included global architecture, restricted vaccine supply chains and limited availability of donations promised by donors through the COVAX facility for low- and middle-income countries by the end of 2021 [21].

Perceptions about the COVID-19 vaccine were mainly positive in our study. There was a high willingness among the unvaccinated to get vaccinated across three of our five sites. While some COVID-19 misinformation was reported, the proportion of participants reporting misinformation was low, with few HCPs believing that COVID-19 is a conspiracy or bioweapon; that people of African descent are immune to COVID-19; that people have died from taking the vaccine; and that the COVID-19 vaccine was unnecessary. Many in the study correctly believed that the side effects of the COVID-19 vaccine are usually mild and temporary. The low levels of misinformation overall could have contributed to lower levels of hesitancy among HCPs.

Most of the HCP who had not been vaccinated indicated a willingness to take the COVID-19 vaccine. However, in Tanzania among the unvaccinated, willingness to take the COVID-19 vaccination was low. The main reasons provided for being unwilling to take the vaccine included perceptions that the vaccine was not effective, concerns about side effects and the belief that the vaccine was developed too fast. The concerns about side effects and safety have also been reported in other studies from similar settings and among the general population [24,30,31].

Beliefs regarding the safety and effectiveness of the COVID-19 vaccine are important considerations for HCP uptake. The majority of HCP in this study believed that COVID-19 vaccines are safe and effective in preventing the disease. In this study, HCP rightly perceived a high risk of exposure to COVID-19 across all countries. In contrast, most of the HCP believed that there is insufficient evidence that the COVID-19 vaccine prevents the occurrence and spread of COVID-19. This observation could reflect sentiments that although Europe, America and Asia had greater availability of vaccines, their reported declines in morbidity lagged as vaccine rollout continued in these contexts. Finally, there were concerns about how quickly the vaccine had been developed in this study since drug development usually takes many years to go through the various pre-clinical and clinical stages. In addition, at least a quarter of respondents indicated that the origin of the vaccine influenced their willingness to take COVID-19 vaccines, and a similar proportion indicated that if a vaccine were developed and tested in Africa, this would increase their willingness to take it. Potential concerns about vaccine safety and efficacy may be partially addressed in the future by encouraging local production of vaccines on the African continent.

The differences between our findings and those from previous studies may be because we included 5 countries in our study and have a larger sample size and variability of responses than previous studies. Our findings can account for differences in contextual factors across the countries and determine overall influential factors affecting vaccine hesitancy across countries.

Given the implications for health outcomes, addressing COVID-19 hesitancy is of critical importance. Current efforts on the African continent to address this barrier to vaccine uptake could be effective if they utilize a multi-pronged approach, that includes community awareness campaigns, extensive stakeholder engagement, various communication approaches and updating government health policies and guidelines [32]. A study in South Africa found that financial incentives such as cash vouchers can increase vaccination rates [33]. Other strategies shown to be effective include mass vaccination campaigns in Zambia and Niger [34]. Additionally, approaches such as integrating COVID-19 vaccination into routine immunization programs and primary health services could be effective [34].

Our study had limitations. Being a telephone survey, it would not have been equally available to all HCP, and it was affected by non-response in some countries. We instituted replacements using randomly allocated individuals from existing sampling frames and HCP lists to mitigate against this. Additionally, 4 of our 5 study sites were urban, and therefore findings may not be attributable to HCPs in rural areas. We defined COVID-19 hesitancy by classifying responses as, definitely not getting the vaccine, maybe, unsure, or undecided on whether to get the COVID-19 vaccine were it available. This decision was made based on existing literature [10] and to ensure consistency across our studies. However, we recognize that these responses may represent individuals with different characteristics that require different interpretations. We also classified all HCP who got the COVID-19 vaccine as not hesitant. This may have resulted in the study underestimating hesitancy among HCP as some would have been mandated to get a vaccine by their employers. However, to our knowledge, none of the countries had a mandate that all HCP should be COVID-19 vaccinated. This is an issue that requires further scrutiny in other studies that collected information on this. Finally, our study was exploratory and descriptive. Future studies in the region should fully explore health behavior models that may explain behaviors around the uptake of COVID-19 vaccines by HCP. This will provide sufficient information to help inform policies and interventions to address this issue.

Our study findings have implications for the current COVID-19 vaccination efforts in countries where vaccine hesitancy was higher than in others and also have implications for future public health emergencies. There is a need to address higher levels of vaccine hesitancy among HCP in Burkina Faso and Tanzania, which are probably reflective of levels of hesitancy in the general population. The root causes of vaccine hesitancy have to be understood and addressed across countries if we are to avoid future challenges in future public health emergencies. Further, studies that are nationally representative including qualitative studies in this region may provide clarity on the reasons for vaccine hesitancy and can identify opportunities to address this problem. Further, the study uncovers differences in perception of risk by profession and possible information asymmetry within different levels of HCP. This may be addressed by providing on-the-job training and refresher courses for lower-level HCP who may have limited access to up-to-date information on COVID-19.

In conclusion, in this study, we found low COVID-19 vaccine hesitancy across most countries, with higher vaccine hesitancy among HCP in Burkina Faso and Tanzania. We also found that factors associated with hesitancy include age, occupation, beliefs about vaccine effectiveness and endorsement by global public health institutions. Efforts to increase vaccine uptake will have to address vaccine hesitancy by tackling information and knowledge gaps among different cadres of HCP, along with efforts to increase vaccine supply.

## Supporting information

**S1 Fig. Unvaccinated health care provide response on "If COVID vaccine is available, I will get it" in 5 countries in sub-Saharan Africa.**
(TIF)

**S2 Fig. Commonly prescribed treatments for COVID-19 in 5 countries in sub-Saharan Africa.**
(TIF)

**S1 Table. Proportion of participants participating in survey rounds.**
(DOCX)

**S1 Text. ARISE COVID19 healthcare provider survey.**
(PDF)

## Acknowledgments

We thank all study participants and data collectors for contributing to this study. The survey team in Ghana appreciates the support from the Kintampo Health Research Centre of Ghana Health Service and the community leadership of Kintampo North Municipality and Kintampo South District. We acknowledge institutional support from Harvard T.H. Chan School of Public Health, Boston, MA; Harvard University Center for African Studies, Boston, MA; Heidelberg Institute of Global Health, Germany and the George Washington University Milken Institute of Public Health, Washington, DC who provided support for the work.

## Author contributions

**Conceptualization:** Isabel Madzorera, Livesy Naafoe Abokyi, Dongqing Wang, Ourohiré Millogo, Nega Assefa, Angela Chukwu, Bruno Lankoande, Elena C. Hemler, Abbas Ismail, Sulemana Abubakari, Kwaku Poku Asante, Yemane Berhane, Japhet Killewo, Ali Sie, Abdramane Soura, Mary Mwanyika-Sando, Said Vuai, Emily Smith, Till Baernighausen, Raji Tajudeen, Wafaie W. Fawzi.

**Data curation:** Isabel Madzorera, Edward Apraku, Temesgen Azemraw, Valentin Boudo, Christabel James, Frank Mapendo, Firehiwot Workneh.

**Formal analysis:** Isabel Madzorera, Livesy Naafoe Abokyi.

**Funding acquisition:** Emily Smith, Till Baernighausen.

**Investigation:** Wafaie W. Fawzi.

**Methodology:** Isabel Madzorera, Dongqing Wang, Ourohiré Millogo, Nega Assefa, Angela Chukwu, Firehiwot Workneh, Bruno Lankoande, Sulemana Abubakari, Kwaku Poku Asante, Yemane Berhane, Japhet Killewo, Ayoade Oduola, Ali Sie, Abdramane Soura, Said Vuai, Till Baernighausen, Wafaie W. Fawzi.

**Resources:** Emily Smith, Wafaie W. Fawzi.

**Writing – original draft:** Isabel Madzorera, Livesy Naafoe Abokyi.

**Writing – review & editing:** Isabel Madzorera, Livesy Naafoe Abokyi, Edward Apraku, Temesgen Azemraw, Valentin Boudo, Christabel James, Dongqing Wang, Frank Mapendo, Ourohiré Millogo, Nega Assefa, Angela Chukwu, Firehiwot Workneh, Bruno Lankoande, Elena C. Hemler, Abbas Ismail, Sulemana Abubakari, Kwaku Poku Asante, Yemane Berhane, Japhet Killewo, Ayoade Oduola, Ali Sie, Abdramane Soura, Mary Mwanyika-Sando, Said Vuai, Emily Smith, Till Baernighausen, Raji Tajudeen, Wafaie W. Fawzi.

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
