## [Decision Letter · Decision Letter 0]

9 Jan 2024

PGPH-D-23-01707

Perceptions and predictors of COVID-19 vaccine hesitancy among healthcare providers across five countries in sub-Saharan Africa

Dear Dr. Madzorera,

Thank you for submitting your manuscript to PLOS Global Public Health. After careful consideration, we feel that it has merit but does not fully meet PLOS Global Public Health’s publication criteria as it currently stands. Therefore, we invite you to submit a revised version of the manuscript that addresses the points raised during the review process.

We look forward to receiving your revised manuscript.

Kind regards,

Vanessa Carels

Staff Editor

Journal Requirements:

2. We have noticed that you have uploaded Supporting Information files, but you have not included a list of legends. Please add a full list of legends for your Supporting Information files after the references list.

Additional Editor Comments (if provided):

Reviewers' comments:

Reviewer's Responses to Questions

**Comments to the Author**

1. Does this manuscript meet PLOS Global Public Health’s publication criteria ? Is the manuscript technically sound, and do the data support the conclusions? The manuscript must describe methodologically and ethically rigorous research with conclusions that are appropriately drawn based on the data presented.

Reviewer #1: Yes

Reviewer #2: Yes

2. Has the statistical analysis been performed appropriately and rigorously?

Reviewer #1: Yes

Reviewer #2: Yes

3. Have the authors made all data underlying the findings in their manuscript fully available (please refer to the Data Availability Statement at the start of the manuscript PDF file)?

Reviewer #1: Yes

Reviewer #2: Yes

4. Is the manuscript presented in an intelligible fashion and written in standard English?

Reviewer #1: Yes

Reviewer #2: Yes

5. Review Comments to the Author

Reviewer #1: Thank you for the opportunity to review this interesting and well-written paper. The paper describes a cross-sectional survey of healthcare providers in Burkina Faso, Tanzania, Ethiopia, Ghana, and Nigeria to assess their perceptions and predictors of their COVID-19 vaccination uptake behaviors and intentions.

The study introduction is framed well within the relevant existing literature of COVID-19 vaccine perceptions and uptake across countries in Africa. My overall view is that the authors have conducted an interesting study that addresses an important global health issue.

It is my opinion that this study offers important insights on in a geographical area that has been comparatively overlooked in the literature. The manuscript is well written and the proposed hypotheses and analyses are logical and compelling. I have only a few comments and suggestions for the authors.

Abstract

- The fourth sentence (beginning line 87), should contain information on the time period (year/months) that the survey was conducted.

Introduction

- Please update the case numbers since March 2022.

Methods

- I am not familiar with the role of pharmacists in sub-Saharan African countries, but it is surprising that they do not have any role in delivering COVID-19 medical related services (i.e., vaccinations, testing) – could the authors provide some reference or supporting material for this statement? (line 171)

- “…to meet sample requirements” (line 185). Was a sample size calculation performed? Either way, this should be discussed.

- “...presented in other manuscripts” (line 197). Please provide references in text.

- I know the authors provide references to two papers (19,20) which contain further details on the ARISE network, but it would be valuable to have a brief overview of the number of survey rounds and when they occurred, this could even be in a timeline as a figure to save adding words. Or state explicitly if there are no rounds other than 1 and 2.

- Could the authors provide a direct link to the survey questions. I followed the link on lines 166-167 but could not access them from the document.

- The authors classified hesitancy as unvaccinated respondents who said they would definitely not, would maybe, were undecided, and were unsure. While the latter three responses indeed correspond to hesitancy, the first response seemingly better represents outright vaccine refusal (Bedford et al., 2018). Did the authors explores whether their findings differed when separating out these two groups? In the discussion the authors mention they grouped together for “ease of data analysis” (line 495), which is not particularly compelling given the potential insights that may come from exploring these two groups separately.

Discussion

- The first line of the discussion should include the time period (year/months) of the survey.

- While the authors reflect on the factors associated with vaccine hesitancy (e.g., lines 424-425) and the patterns of findings across countries, the discussion section would benefit greatly from discussion of existing efforts to address these barriers to vaccine uptake (e.g., mandates, messaging strategies, incentives). If there are no studies specifically from the relevant geographic regions the authors should highlight work done in other countries to address similar barriers to vaccine uptake and reflect on the need for their replication in this context.

Reviewer #2: The aim of this study was to describe the attitudes and behaviours of healthcare providers in 5 countries (Ethiopia, Nigeria, Burkina Faso, Ghana and Tanzania) and the factors associated with COVID-19 vaccine hesitancy. Using a cross-sectional study, the authors present a picture of the 2nd half of 2021 in sub-Saharan Africa. Many descriptive data are presented by the authors.

Abstract

Lines 98-99: this sentence is repetitive of the one in lines 92-93. It can be removed.

“We found higher vaccine hesitancy among HCP in Burkina Faso and Tanzania.”

Statistical analysis

Line 212: It would be good to provide a table or document with all the variables, both potential confounder questions and the various questions on health behaviours or attitudes.

The choice of analysis to predict the behaviour of HCPs is not entirely clear (rational).

Results

There are several sections of Results that could be reduced in length. Repeating values in tables is not helpful for readers.

Table 2

Lines 303-305: What the n-value and % represent isn't clear. Are they the proportion of participants who answered "Yes"? It's not entirely clear what is presented in the tables in general concerning the questions on "Knowledge about COVID-19 vaccines" and "COVID-19 Prevention practices".

Table 3

Lines 358-360: Could "Vaccine hesitancy" be replaced by "vaccine hesitant" since we're talking about HCPs?

Table 4

Lines 394-396: It is important to list the variables of the adjusted model under the table. “All variables included in the final model are shown in the adjusted model.”

Discussion

There are several questions the authors could address in the discussion:

- What are the health policies of different countries regarding vaccination (and other vaccines)?

- What information HCPs had received during that period?

Lines 493-494: The authors need to differentiate between vaccine hesitance and refusal (as present in the MacDonald 2015 article - Figure 1). Your choice of data to define hesitancy is problematic since you also include people who refuse, i.e. who are not hesitant. This should be reviewed.

“We defined COVID-19 hesitancy by classifying responses as, definitely not getting the vaccine, maybe, unsure, or undecided on whether to get the COVID-19 vaccine were it available.”

6. PLOS authors have the option to publish the peer review history of their article (what does this mean? ). If published, this will include your full peer review and any attached files.

**Do you want your identity to be public for this peer review?** For information about this choice, including consent withdrawal, please see our Privacy Policy .

Reviewer #1: No

Reviewer #2: No

---

## [Decision Letter · Decision Letter 1]

25 Jun 2024

PGPH-D-23-01707R1

Perceptions and predictors of COVID-19 vaccine hesitancy among healthcare providers across five countries in sub-Saharan Africa

Dear Dr. Madzorera,

Thank you for submitting your manuscript to PLOS Global Public Health. After careful consideration, we feel that it has merit but does not fully meet PLOS Global Public Health’s publication criteria as it currently stands. Therefore, we invite you to submit a revised version of the manuscript that addresses the points raised during the review process.

We look forward to receiving your revised manuscript.

Kind regards,

Everton Falcão de Oliveira, Ph.D

Academic Editor

Journal Requirements:

Additional Editor Comments (if provided):

Reviewers' comments:

Reviewer's Responses to Questions

**Comments to the Author**

1. If the authors have adequately addressed your comments raised in a previous round of review and you feel that this manuscript is now acceptable for publication, you may indicate that here to bypass the “Comments to the Author” section, enter your conflict of interest statement in the “Confidential to Editor” section, and submit your "Accept" recommendation.

Reviewer #3: All comments have been addressed

Reviewer #4: All comments have been addressed

2. Does this manuscript meet PLOS Global Public Health’s publication criteria ? Is the manuscript technically sound, and do the data support the conclusions? The manuscript must describe methodologically and ethically rigorous research with conclusions that are appropriately drawn based on the data presented.

Reviewer #3: Yes

Reviewer #4: Yes

3. Has the statistical analysis been performed appropriately and rigorously?

Reviewer #3: Yes

Reviewer #4: Yes

4. Have the authors made all data underlying the findings in their manuscript fully available (please refer to the Data Availability Statement at the start of the manuscript PDF file)?

Reviewer #3: Yes

Reviewer #4: Yes

5. Is the manuscript presented in an intelligible fashion and written in standard English?

Reviewer #3: Yes

Reviewer #4: Yes

6. Review Comments to the Author

Reviewer #3: Introduction

I appreciate the opportunity to review such a relevant manuscript, considering the challenge of collecting data on vaccine hesitancy and its implications. The research describes the perceptions and predictors of healthcare professionals in five sub-Saharan African countries in order to analyze and discuss the factors associated with vaccine hesitancy. The authors address important questions and have shown a willingness to analyze the implications and particularities of each country.

Line 123: In this part, the term "herd immunity" is used; I suggest replacing it with the term "community immunity". There is a global trend to avoid using terms that refer to animals, thus preventing their pejorative use in other contexts and popularizing the term community immunity.

Line 128-129: The phrase "[...] the management AND control of COVID-19 AND have a high risk of getting infected." Please replace the first "and" with a comma ",".

Material and Methods

Lines 188-196: As I read through, I had a question, particularly at this point: was there a criterion for selecting participants for the second round? Were the participants in the second round those who indicated vaccine hesitancy? I reviewed the methods section, and this doubt remained. Please point me to the exact description within the manuscript.

Results

I really appreciated the tables; they were well-organized and easy to understand.

Table 2: I suggest splitting the table into two parts. Separate the "Frequency of availability of COVID-19 related services" from the "Knowledge about the vaccine among professionals". This would facilitate visualization and referencing of the tables throughout the results description.

Page 20: It is indicating "Figure 3: Frequent reasons (%) for not getting COVID-19 vaccination in 5 countries in sub-Saharan Africa," but Figure 2 shows the same title. Please correct the numbering.

Discussion

Some questions were answered in the discussion reading, and I appreciated how the authors addressed these findings, such as vaccinated healthcare professionals may have hesitated at some point and maintained their belief regarding COVID-19 vaccination and checking if there is any mandatory vaccination of this class.

I consider it pertinent that you can address more incisively the fact that nurses are among the most hesitant class along with what already exists in the literature, as nurses are on the front line and we can say that during the COVID-19 pandemic they had contact with infected people for some period at any level of health care.

It was extremely important to address the study limitations, as this demonstrates that the researchers involved are attentive to data quality. I hope the study can generate a suitable impact, especially within a profession that holds authority in the healthcare field and is capable of educating health users.

Reviewer #4: The paper provides important insights on the perception of HCP in SSA regarding COVID-19 immunizers. The findings are highly important for the working field. However, a few points still need to be addressed in the following section:

Methodology:

(Lines 153 and 173) The methodology section could be better structured to enhance the reader's comprehension. The subsection "study design" is presented in line 153 and repeated in line 173.

(Line 159) Consider changing "study participants" to "eligible participants."

(Issue with clarity of the study sampling throughout the methodology section) The authors repeatedly mention that the study was divided into two rounds. Presenting briefly the main purpose of the survey, the purpose of each round, a brief explanation of the data collected in both rounds, and explaining that the paper presents data from the second round and the reasons for this at the beginning of the methodology section would make it easier to comprehend the following subsections.

(Lines 171-172) The justification for the exclusion of dentists and pharmacists (and other healthcare personnel besides nurses and doctors) is still insufficient. Dentists are at high risk of acquiring COVID-19, and pharmacists are information providers for the population. The authors justify their exclusion by stating these professionals are unlikely to deliver COVID-19 medical services, but is that accurate? Clarifying that the study addressed hesitancy only in nurses and medical professionals such as physicians because those were the professionals currently available, or better clarifying the reasons for this choice, would be more suitable.

(Line 173) The study sampling is quite unclear; the authors point out limitations but not how the sampling was conducted.

(Lines 204-207) Even though the authors point to this as a limitation later in the paper, the concept of vaccination hesitancy is an issue throughout the manuscript. The concept referenced is the vaccination scope of the WHO from 2015, which seems more related to the willingness to get vaccinated, as HCP who were vaccinated were classified as non-hesitant. However, the vaccination hesitancy scope of 2015 includes acceptance with doubts, which was not assessed in the paper. Additionally, refusal is also included in the scope and was excluded from further analysis. The new concept from 2022 ("Vaccine hesitancy” is part of the Motivation domain and defined as a motivational state of being conflicted about, or opposed to, getting vaccinated; this includes intentions and willingness") could be assessed, but classifying HCP who got the vaccine with doubts would still be an issue. Maybe instead of hesitancy, assess willingness in non-vaccinated HCP to minimize the conceptual issue, as throughout the discussion, the authors mention willingness.

(Collected data) As the study collected many variables, I suggest pointing out the variables collected as they are presented in the subsections of the results (for example: socio-demographic variables: follow variables presented in Table 1, and so on) to make it easier to comprehend the results section and the collected variables.

Lastly, the results are concerning to the study field, and the study is extremely interesting and important. Just a few adjustments are still needed. I congratulate the authors on the great work!

7. PLOS authors have the option to publish the peer review history of their article (what does this mean? ). If published, this will include your full peer review and any attached files.

**Do you want your identity to be public for this peer review?** For information about this choice, including consent withdrawal, please see our Privacy Policy .

Reviewer #3: No

Reviewer #4: No

---

## [Decision Letter · Decision Letter 2]

29 Oct 2024

Perceptions and predictors of COVID-19 vaccine hesitancy among healthcare providers across five countries in sub-Saharan Africa

PGPH-D-23-01707R2

Dear Dr Madzorera,

We are pleased to inform you that your manuscript 'Perceptions and predictors of COVID-19 vaccine hesitancy among healthcare providers across five countries in sub-Saharan Africa' has been provisionally accepted for publication in PLOS Global Public Health.

Best regards,

Everton Falcão de Oliveira, Ph.D

Academic Editor

Reviewer Comments (if any, and for reference):

Reviewer's Responses to Questions

**Comments to the Author**

1. If the authors have adequately addressed your comments raised in a previous round of review and you feel that this manuscript is now acceptable for publication, you may indicate that here to bypass the “Comments to the Author” section, enter your conflict of interest statement in the “Confidential to Editor” section, and submit your "Accept" recommendation.

Reviewer #3: All comments have been addressed

Reviewer #4: All comments have been addressed

2. Does this manuscript meet PLOS Global Public Health’s publication criteria ? Is the manuscript technically sound, and do the data support the conclusions? The manuscript must describe methodologically and ethically rigorous research with conclusions that are appropriately drawn based on the data presented.

Reviewer #3: (No Response)

Reviewer #4: (No Response)

3. Has the statistical analysis been performed appropriately and rigorously?

Reviewer #3: (No Response)

Reviewer #4: Yes

4. Have the authors made all data underlying the findings in their manuscript fully available (please refer to the Data Availability Statement at the start of the manuscript PDF file)?

Reviewer #3: (No Response)

Reviewer #4: Yes

5. Is the manuscript presented in an intelligible fashion and written in standard English?

Reviewer #3: (No Response)

Reviewer #4: Yes

6. Review Comments to the Author

Reviewer #3: (No Response)

Reviewer #4: The methodology section has been greatly improved, it is comprehensible the limitation of the data and the authors are to be congratulated for the great work.

7. PLOS authors have the option to publish the peer review history of their article (what does this mean? ). If published, this will include your full peer review and any attached files.

**Do you want your identity to be public for this peer review?** For information about this choice, including consent withdrawal, please see our Privacy Policy .

Reviewer #3: No

Reviewer #4: No
